# Viscoelastic, Thermal, and Mechanical Properties of Melt-Processed Poly (ε-Caprolactone) (PCL)/Hydroxyapatite (HAP) Composites

**DOI:** 10.3390/ma15010104

**Published:** 2021-12-24

**Authors:** Mpho Phillip Motloung, Tladi Gideon Mofokeng, Suprakas Sinha Ray

**Affiliations:** 1Centre for Nanostructures and Advanced Materials, DSI-CSIR Nanotechnology Innovation Centre, Council for Scientific and Industrial Research, Pretoria 0001, South Africa; MMotloung@csir.co.za (M.P.M.); tmofokeng@csir.co.za (T.G.M.); 2Department of Chemical Sciences, University of Johannesburg, Johannesburg 2028, South Africa

**Keywords:** morphology, tensile properties, flow properties, crystallinity

## Abstract

Poly (ε-caprolactone) (PCL)/hydroxyapatite (HAP) composites represent a novel material with desired properties for various applications. In this work, PCL/HAP composites at low loadings were developed through melt-extrusion processing. The effects of HAP loading on viscoelastic, thermal, structural, and mechanical properties of PCL were examined. The morphological analysis revealed better dispersion of HAP at low loadings, while aggregation was noticed at high concentrations. The complex viscosity of the prepared composites increased with increasing concentration of HAP. In addition, a significant decrease in crystallinity was observed upon increase in HAP loading. However, the elongation at break increased with increasing the concentration of HAP, probably due to a decrease in crystallinity. The onset thermal degradation temperature of PCL was enhanced at low concentrations of HAP, whereas a decrease was observed at high loading. Overall, different degrees of HAP dispersion resulted into specific property improvement.

## 1. Introduction

In recent years, research effort has been diverted into developing environmentally benign materials. In this regard, the biodegradable polymers have received enormous interest to replace the conventional polymers, which are non-environmentally friendly [1,2]. Among many different biodegradable polymers, poly (ε-caprolactone) (PCL) is a biodegradable polymer derived from fossil fuels, which has garnered attention in both academia and industrial sector thanks to its biodegradability and biocompatibility. PCL is a ductile semi-crystalline polymer with low melt processing temperatures [1,3]. Due to its ductility, PCL is commonly used as a toughening agent for brittle polymers such as polylactic acid (PLA) and poly (hydroxyalkatone) (PHA) [4,5]. Previous studies have demonstrated the potential applications of PCL in various sectors such as biomedical [6], packaging [7], textile [8], and others.

Polymer composites represent the novel materials with specific functionality imparted by both the filler and polymer matrix. Usually, the intrinsic properties of a polymer can be altered by inclusion of filler in a polymer. Numerous filler materials have been incorporated into PCL to enhance its certain properties such as conductivity [9,10], barrier [7,11], sensing [12], and biological activity [13,14]. Amongst many different fillers, hydroxyapatite (HAP) has received a significant interest mostly due to its exceptional properties. HAP is an inorganic bio-filler, which resembles the properties of bone; hence, it is mainly used in biomedical applications due to its biocompatibility [15,16,17]. Several studies have developed PCL/HAP composite scaffolds for biomedical applications [13,14,18]. However, HAP in combination with other materials such as urea, can be used for a control of fertilizer release in soil [19,20]. HAP is a phosphorus-containing compound; hence, it is favoured for fertilizer release. Further, HAP has been used as a compatibilizer in biodegradable polymer blends [17]. Many reported studies have prepared PCL/HAP composites through solvent casting [14,21,22,23]. For example, Hao et al. [21] prepared PCL/HAP composites with varying concentrations of HAP (7.4–28.6 wt.%) through a solvent-casting process using dimethylformamide (DMF) as a solvent. The prepared composites showed higher crystallization temperatures (T_c_) compared to neat PCL, indicating the nucleating effect of HAP in PCL. Further, the tensile modulus of the composites showed increasing trends when increasing the amount of HAP; whereas, the yield stress showed little dependence on the concentration of HAP suggesting better dispersion of the filler in the composites. Chuenjitkuntaworn et al. [22] developed PCL/HAP scaffolds for bone tissue engineering through solvent casting and particulate leaching techniques. The compressive modulus of the PCL/HAP composites increased with increasing the concentration of HAP, suggesting the reinforcing effect of HAP in PCL. In addition, the PCL/HAP scaffolds showed better adhesion and proliferation of human bone cells. It is worth mentioning that solvent-based processes are not favourable because of health issues related to use of toxic organic solvents and their evaporation polluting the environment. In addition, it is challenging to scale-up the solvent-based process of preparing polymer composites. However, the 3D printed scaffolds can be alternative to solvent-processed scaffolds. Gerdes et al. [24] and Zavřel et al. [25] prepared 3D printed PCL/HAP scaffolds for bone tissue engineering. 3D scaffolds can easily be prepared at large scale since the process is free of solvents, and the scaffolds with different architectures can be prepared.

Melt processing of polymer composites is crucial for a large-scale production since the process is continuous and free of solvents. However, the flow properties of the material need to be understood. This is important especially for continuous process applications such as film blowing and melt spinning, which require adequate melt strength for better processability and good dispersion and compatibility between polymer and filler for enhanced material properties. Post-melt processing of polymer composites such as melt spinning can be an alternative to the most used processes such as electrospinning of PCL/HAP composites. Nonetheless, the extent of filler dispersion, interactions, and the overall properties of the melt-processed composites require exhaustive understanding. It is worth mentioning that HAP is usually added at high concentrations and the prepared composites are intended to be applied for medical purposes. Generally, filler particles tend to agglomerate at high amounts and pose difficulties in material processability and harm the resulting properties. However, Akhbar et al. [26] prepared the PCL/HAP composites containing high concentrations of filler. In their study, the authors employed ultrasonic wave to assist with the processability of the composites during extrusion. Though ultrasonic wave assisted strategy shows to be feasible to process the composites at high loading, the overall process might become expensive, due to additional use of ultrasonic wave technique. Most recently, Backes et al. [27] prepared PCL/HAP composites using Haake torque rheometer. The concentration of HAP in PCL was 5, 10, and 25 wt.%. The gel permeation chromatography (GPC) analysis indicated a decrease in the number average molecular weight (M_n_) and weight average molecular weight (M_w_) of PCL with increasing the amount of HAP. The reduction in M_n_ and M_w_ was associated with accelerated polymer chain scission by calcium phosphate such as HAP. At 25 wt.% HAP, the M_n_ reduced by approximately by 29% with respect to neat PCL. Generally, a reduction in molecular weight has detrimental effects on the properties of a polymer, which includes increase in melt flow index (MFI) and thus subsequent reduction in melt strength. Thus, it is important to minimize the amount of HAP in PCL composites to maintain or avoid huge reduction in molecular weight. For other applications, which do not necessarily need high loading of HAP, the composites can still be prepared and in such cases, the HAP will act as reinforcing agent and enhance some of the material characteristics at low amounts. In this study, compared to other studies which used high loadings of HAP, the effects of HAP nanoparticles incorporated at low concentrations on the properties of PCL were investigated. An insight on the resulting properties of PCL/HAP composites, particularly, the rheological, thermal, and mechanical behaviours is provided. The composites were prepared through melt mixing; a process that can be scaled-up to industry level and is suitable for other different process applications such as melt spinning, film blowing, etc. The concentration of HAP was varied from 1 to 7 wt.%, which is low compared to other studies that used high concentrations of 20 [28] and 60 wt.% [29] of HAP, possibly due to target application of the final composite in bone regeneration.

## 2. Materials and Methods

### 2.1. Materials

HAP (size < 200 nm, BET-according to a supplier) was purchased from Sigma Aldrich, South Africa. PCL (Celgreen PH-7) used in this study was obtained from Daicel Chemical Industries Co., Ltd., Tokyo, Japan. The material has melt-flow rate of 6 ± 0.4 g/10 min at 100 °C and weight of 2.16 kg, as determined according to ISO standard 1133 using CEAST MFM Multiweight instrument.

### 2.2. Sample Preparation

The PCL pellets were dried for ~24 h at 30 °C in vacuum oven prior melt processing to remove moisture. The PCL/HAP nanocomposites, including neat PCL, were melt processed in twin-screw extruder (Process 11, Thermo Scientific, Waltham, MA, USA) with L/D of 40. The materials were processed at temperature of 100, 100, 100, 100, 100, 60, 40 (die to feed zone). The extruded polymer strands were cooled in water bath, followed by palletisation, then drying at 35 °C for 24 h. The concentration of HAP was varied from 1 to 7 wt.% with respect to PCL, and the sample compositions were as follows: PCL0 (neat PCL), PCL1 (PCL/1 wt.% HAP), PCL3 (PCL/3 wt.% HAP), PCL5 (PCL/5 wt.% HAP), and PCL7 (PCL/7 wt.% HAP).

### 2.3. Characterization

#### 2.3.1. Scanning Electron Microscopy

The morphologies of cryo-fractured PCL/HAP composites, sputter coated with carbon, were determined using scanning electron microscopy (SEM) (JSM-7500, JEOL, Tokyo, Japan) at voltage of 3.0 kV.

#### 2.3.2. Polarized Optical Microscopy

The level of HAP dispersion in PCL was investigated using polarized optical microscopy (POM). The samples were prepared into thin films, sandwiched between the glass coverslips, and then heated to 100 °C on the THMS hot stage (Linkam Scientific Instruments, Ltd., Surrey, KT, UK) at heating rate of 20 °C/min. The samples were held isothermal at 100 °C for 10 min to ensure complete melting and the images were acquired using Carl Zeiss imager, Carl Zeiss, Germany).

#### 2.3.3. Chemical Structures

The chemical structures of neat PCL, HAP, and PCL/HAP composites were determined using attenuated total reflectance (ATR) Fourier transform infrared (FTIR) spectroscopy (Perkin-Elmer Spectrum 100 spectrometer, Branford, CT, USA) at the wavelength range of 550 to 4000 cm^−1^ at resolution of 4 cm^−1^.

#### 2.3.4. Crystalline Structures

The crystalline structures of neat PCL, HAP, and PCL/HAP composites were determined through X-ray diffraction (XRD) X’Pert PRO diffractometer (PAN analytical, EA Almelo, Netherlands) producing Cu Kα radiation (λ = 1.54 nm), operated at 45 kV and 40 mA.

#### 2.3.5. Rheological Properties

The flow characteristics of the prepared samples were obtained using rheometer (Physica MCR 501, Anton Paar, Garz, Austria) operated at a temperature of 100 °C using parallel plates (25 mm) configuration, The frequency sweep tests were carried, with pre-determined strain amplitude of 1%.

#### 2.3.6. Thermal Properties

Thermal degradation properties of the PCL, HAP and PCL/HAP composites weighing approximately 8 mg were determined using PerkinElmer Pyris 1 TGA. The samples were heated from 30 to 800 °C in the steel pans, at a heating rate of 10 °C/min under nitrogen environment (flow rate = 20 mL/min).

The thermal transitions (melting and crystallization temperatures, including the melting enthalpies (ΔH_m_)) occurring in PCL and PCL/HAP composites were investigated using differential scanning calorimetry (DSC) (DSC-Q2000, TA Instruments, New Castle, DE, USA). The sample mass of approximately 5 mg was used for testing. The tests were run from −20 to 100 °C at a heating rate of 10 °C/min (both melting and cooling) under nitrogen environment with flow rate of 25 mL/min. The transitions occurring during melting were determined from the second heat curve.

#### 2.3.7. Mechanical Properties

The elongation at break and ultimate tensile strength were determine through tensile testing using Instron 5966 tester (Instron Engineering Corporation, Norwood, MA, USA) with load cell of 10 kN.

## 3. Results and Discussion

### 3.1. Surface Morphology and Chemical Structures

The SEM micrograph of HAP particles is shown in Figure 1a. The particles are spherical and have average diameter of 53 ± 7 nm. The dispersion of HAP in PCL was investigated through SEM and the results are depicted in Figure 1b–e. Generally, nanoparticles disperse better at low concentrations, whereas agglomerations are mostly noticed at higher filler concentrations. In the current case, similar behaviour is also observed where HAP shows better dispersion at 1 and 3 wt.%, while agglomerations of the particles could be seen at high concentrations of 5 and 7 wt.%. The dispersion of HAP can be enhanced through optimization of processing conditions such as shear rate, which can effectively aid in breaking the agglomerates and enhance the dispersion of filler.

Figure 2 shows the optical microscopy of the HAP particles in the melt state, and it could be noticed that the results are in accordance with the SEM results. The agglomeration of the particles could clearly be observed at high HAP loading, whereas finer dispersion of the particles is discernible at low amounts. The specific chemical interactions between HAP and PCL cannot be observed from the microscopy analysis. However, the FTIR analysis could enlighten us on how both PCL and HAP interact during melt processing. Chemical interactions in biphasic materials such as polymer composites or blends are accompanied by formation/breaking of specific bonds and result into disappearance/appearance of new bonds on the FTIR. The chemical structures of PCL and PCL/HAP composites are depicted in Figure 3. PCL is characterised by strong carbonyl (–C=O) stretching peak at 1720 cm^−1^. The other stretching peaks noticed at 2900 and 2800 cm^−1^ are attributed to asymmetric and symmetric –CH_2_ stretching, respectively. The other peaks could be noticed at 1293 and 1240 cm^−1^ and are attributed to C–O and C–C stretching in the crystalline phase and asymmetric C–O–C stretching, respectively [30,31,32]. On the other hand, HAP is characterized by a phosphate-stretching bands (P–O) at 1099, 990, 1099 cm^−1^. The other peaks noticed at 590, 560, and 549 cm^−1^ are attributed to bending mode of P–O–P group [33,34]. The inclusion of HAP did not have any effect on the chemical structure of PCL at low amounts. However, with increase in HAP loading, the appearance of new peak could be noticed at 610 cm^−1^ and it could be attributed to the presence of HAP in PCL.

### 3.2. Viscoelastic Properties

#### 3.2.1. Complex Viscosity

The evolution of complex viscosity as a function of angular frequency is illustrated in Figure 4a. Generally, for linear polymers comprising one phase, the complex viscosity shows two parts; that is, Newtonian behaviour at low angular frequencies and pseudoplastic behaviour at high frequencies. It is apparent from Figure 4a that PCL exhibits Newtonian behaviour at low frequencies (˂10 rad/s), whereas the shear-thinning behaviour is noticed at high frequencies. Similar trend could be observed upon addition of HAP at varying concentrations from 1 to 7 wt.%. The composites showed the independence of complex viscosity at low angular frequencies, albeit the viscosity increased with increasing the concentration of filler, insinuating a restriction on the mobility of PCL chains by HAP nanoparticles. To elucidate on the viscosity of PCL, the complex viscosity modelling was carried to determine the effect of HAP on the relaxation of PCL chains.

#### 3.2.2. Complex Viscosity Modelling

The viscoelastic behaviour observed in the prepared PCL-based composites was explained based on the complex viscosity modelling approach. The generalized complex viscosity equation (Equation (1)) is used to fit the experimental data and it can be expressed as follows [35,36,37]:(1)$$\eta^{*}=k+\eta_{0}{\left[1+{\left(\lambda \omega \right)}^{a}\right]}^{\left(n-1\right)/a}$$
where *η**, *k*, *η_0_*, *λ*, *n*, and *a* represent complex viscosity, intrinsic material property, zero shear viscosity, relaxation time, power law index, and Newtonian transition factor, respectively.

In this Equation (1), depending on the boundaries of *K* and *a*, various models have been proposed and were used in this study to determine the zero-shear viscosity, yield stress, and the relaxation time as a function of filler concentration. The zero-shear viscosity and relaxation times were determined from different models as shown in Equations (2)–(4).

The Carreau–Yasuda model:(2)$$\eta^{*}=\eta_{0}{\left[1+{\left(\lambda \omega \right)}^{a}\right]}^{\left(n-1\right)/a}$$

The Cross model:(3)$$\eta^{*}=\frac{\eta_{0}}{1+{\left(\lambda \omega \right)}^{1-n}}$$

The Berzin model:(4)$$\eta^{*}=\frac{\sigma_{0}}{\omega }+\eta_{0}{\left[1+{\left(\lambda \omega \right)}^{a}\right]}^{\left(n-1\right)/a}$$

Figure 5 shows the Carreau–Yasuda and Cross models used to fit the rheology data for PCL-HAP nanocomposites. It is observed that Carreau–Yasuda model could perfectly predict the complex viscosity of neat PCL and the composites containing up to 5 wt.%, whereas the Cross model was only suitable for composites containing up to 3 wt.% of HAP. Although both models hold for neat polymers, their perfect prediction for complex viscosity composites containing up to 5 and 3 wt.% HAP suggests that the macromolecular topology of PCL was not affected when HAP was incorporated at these concentrations. However, for high amounts of filler (7 wt.%), both models failed to predict the complex viscosity of the materials. The fitting parameters for Carreau–Yasuda and Cross model are summarized in Table 1. The zero-shear viscosity increased with the concentration of HAP in PCL according to both models, implying the formation of filler network in the PCL matrix. The Carreau–Yasuda model indicates that the composites had higher relaxation times than neat PCL, confirming the restriction of PCL chains in the presence of filler. Further, this observation indicates the transition to the pseudo-solid-like behaviour upon increasing the amount of HAP. On the other hand, the Cross-model shows no change in the relaxation time upon increase in filler content. The Berzin model was also used to predict the complex viscosity of PCL composites. It should be noticed that this model includes the strain component (Equation (3)), which is defined as the ratio of yield-stress to angular frequency. Thus, the yield-stress contribution from filler is also considered [35]. However, in this case, it is surprising that the yield strain is zero, which can agree with Cross model that also indicated no change in the relaxation time. As a result, the Berzin model reduces to Equation (2). Based on this observation, it can be inferred that the PCL chains were not confined by the HAP particles; hence, no change in relaxation time and zero yield stress were obtained according to both models.

#### 3.2.3. Storage and Loss Modulus

The dependence of both storage (G′) and loss (G″) moduli on the angular frequency is illustrated in Figure 4b,c. For neat PCL, both G′ and G″ increased according to Maxwell theory of liquids; that is, G′ α ω^2^ and G″ α ω. However, upon loading of HAP particles, the moduli increased over the entire range of angular frequency and followed similar behaviour as neat PCL. Comparing the two moduli, it is evident that the viscous response is dominant over the elastic behaviour (G″ < G′), showing the pseudo-liquid-like behaviour of the samples. Moreover, the increase in G′ with amount of HAP is of almost the same magnitude as of G″, suggesting equal effects of HAP on both viscous and elastic portion of PCL. At high amount of filler (7 wt.%), a clear plateau could be observed at the terminal zone (Figure 4b), which suggests a transition from the pseudo-liquid to pseudo-solid-like behaviour. This observation indicates the formation of 3D network structure formed by filler particles in PCL, especially at high concentrations.

#### 3.2.4. Damping Factor

The transition from liquid to solid-like behaviour can be determined from damping factor (tan δ), which is the ratio of G″/G′. For ideal viscous behaviour, δ approaches 90° and tan δ approaches infinity, whereas the elastic behaviour is noticed when δ = 0° and tan δ is zero. Figure 4d shows the evolution of the damping factor as a function of angular frequency. Neat PCL and composites show an initial increase in damping factor at low angular frequency, and this could be attributed to a decrease in elasticity of PCL according to definition. However, with increase in frequency, a decrease in damping factor with negative slope, indicating a viscous behaviour of the prepared materials was apparent. In addition, with increasing the concentration of HAP, a decrease in damping occurred over the entire range of angular frequency.

### 3.3. Melting and Crystallization Behaviour

Figure 6 demonstrates the melting and crystallization behaviour of the PCL-HAP composites at different loadings. The values for melting and crystallization temperatures, and heat of enthalpy are summarized in Table 2. PCL is characterised by low melting temperatures (≤60 °C) and high crystallinity compared to many other different biopolymers. In the current case, the melting temperature of PCL was observed at 56 °C, although addition of HAP did not show any effects on the melting of PCL. On the other hand, the crystallization temperature of PCL was also not affected as indicated in Table 2. The crystallinity of the prepared composite materials was calculated according to Equation (5).
(5)$$X_{c}=\frac{∆H_{m}}{∆H^{0}{}_{m}\;\left(1-\alpha \right)}\times 100$$
*X*_c_ is percentage crystallinity, Δ*H_m_* is specific melting enthalpy of PCL, Δ*H*^0^*_m_* is specific melting enthalpy for 100% crystalline PCL and its value is 135 g^−1^, α is the weight fraction of HAP filler in PCL [38].

Usually, nanoparticles tend to act as nucleating agents and increase the crystallinity and crystallization temperatures of a polymer; however, that behaviour could not be observed in this case. Looking at the crystallinity of PCL, it did not change significantly with varying the concentration of HAP. A slight decrease in crystallinity could be observed in PCL 7 and this is ascribed to poor dispersion of HAP (Figure 1), which results into hampering of nucleation and crystal growth in PCL.

The XRD is a powerful tool, which can be used to determine the crystallinity of the material and it is complementary to DSC. Figure 7a shows the XRD patterns (with corrected baseline), of PCL and composites containing HAP. PCL is characterized by intense diffraction peaks at 2θ = 21.7 and 23.8°, which correspond to (110) and (200) planes, respectively [26]. The crystalline structure of PCL did not change in the presence of HAP; however, the crystallinity of PCL decreased. The crystallinity is directly proportional to the intensity of the diffraction peaks [39]. Figure 7b shows the changes in the intensity of the main diffraction peaks observed at 2θ = 21.4 and 23.7°. A decrease in the intensity could be observed, especially on diffraction peak at 21.4°. Therefore, it could be inferred that adding HAP decreases the crystallinity of PCL, especially at high concentrations and this agrees with the DSC analysis that also indicated reduction in overall crystallinity of PCL at high HAP loading.

### 3.4. Thermal Stability

The thermal stability of the prepared samples was determined, and the results are shown in Figure 8 and Table 3. Neat PCL and the composites containing up to 3 wt.% HAP showed a single step degradation, with a maximum degradation temperature recorded at around 406 °C. However, at 5 and 7 wt.% loading, the maximum degradation temperature decreased significantly, possibly because of poor dispersion of HAP in PCL. In addition, two-step degradation curves were noticed for composites containing 5 and 7 wt.% HAP (Figure 8b). The first degradation is apparent at 332 °C and it could be attributed to HAP. The effect of HAP loading on the initial degradation of PCL was also evaluated. The 5 % degradation was taken subjectively as an onset degradation and the results are tabulated in Table 3. It is discernible that incorporation of HAP at low concentrations (1 wt.%) significantly enhances the onset thermal degradation temperature of PCL, while a decrease was noticed with increasing the amount of filler. At high concentrations of HAP (5 and 7 wt.%), the onset degradation decreased and this could be due to poor dispersion of HAP in PCL, which decreased the activation energy barrier and hence early degradation. It has been reported that calcium phosphates such as HAP has tendency of inducing polymer chain scission, especially when added at high concentrations [27]. Therefore, this could also explain a noticed decrease in the onset degradation of composites at 5 and 7 wt.% HAP.

### 3.5. Mechanical Properties

The mechanical performance of the prepared PCL/HAP composites was investigated. The stress–strain tests were carried to determine the tensile properties of the samples and the results are depicted in Figure 9, which shows elongation at break (E_b_) and ultimate tensile strength (UTS) as a function of HAP content. PCL is a ductile material with high E_b_ (>300 %) as observed in Figure 9a. With addition of 1 wt.% HAP, the E_b_ did not change indicating no effect on the toughness of PCL. At 3 wt.%, a slight increase in toughness could be noticed. With increase in HAP up to 5 and 7 wt.%, the E_b_ further increased by 15 and 31 % with respect to neat PCL, respectively. Generally, the E_b_ is influenced by several factors including crystallinity, interaction between a polymer and filler, and degree of filler dispersion. Crystallinity is one of the governing factors for overall intrinsic properties of a polymer material. It can be deduced that a decrease in crystallinity contributed to increasing the E_b_ of PCL, though the expectation was that the E_b_ would decrease due to poor dispersion of HAP. However, A decrease in crystallinity as noticed from DSC and XRD suggests less confinement of PCL chains; hence, the increase in the mobility of chains and thus the toughness. Li et al. [29] also observed an increase in E_b_ of electrospun PCL when HAP was introduced at 30 and 60 wt.%. It was concluded that the energy absorbed before the fracture increases in the presence of HAP. The UTS is important as it indicates the maximum stress the material can bear. The UTS of the composites is shown in Figure 9b and addition of HAP resulted into a decrease in UTS, especially at low filler content. However, a slight increase is noticed at high filler content.

## 4. Conclusions

Environmentally benign materials with added functionality are desired for various types of applications. Herein, the effect of HAP loading at low concentrations was investigated. Numerous studies reported incorporated HAP at high amounts, and this can have significant influence on the reduction of molecular weight and properties of a polymer matrix. In this study, PCL/HAP composites were successfully prepared through melt processing technique. It was demonstrated that HAP disperse much better at low concentrations in PCL, whereas agglomerations could be noticed with higher loading of the HAP. The complex viscosity of the prepared composites increased significantly with increasing the amount of HAP in the composites. Increasing the complex viscosity can be an indication of increasing the melt strength of a polymer, and this can be crucial for further processes such as foaming that requires high/moderate melt strength and viscosity for bubble nucleation and growth, and for film blow process, which also requires high melt stability for bubble stability during blowing. On the other hand, the crystallinity of the composites decreased with respect to neat PCL as noticed from the XRD analysis and DSC, especially at high loadings. However, the tensile properties, particularly the E_b_ showed an increasing trend with increasing the HAP concentration. Although one would expect the composites containing high amounts of filler to have poor mechanical performance due to agglomerations noticed, the opposite trend could be noticed and was attributed other factors such as decreasing the crystallinity as noticed from the XRD and DSC analysis. The thermal stabilities, however, showed improvements only at low amount of HAP, possibly due to better dispersion. Overall, a new insight on the properties of melt processed PCL/HAP composites was provided. The developed materials can be used for different applications such as foams and films for packaging application, mulch films for agricultural applications, and textile applications such as short-lived geotextile applications.

## Figures and Tables

**Figure 1 materials-15-00104-f001:**
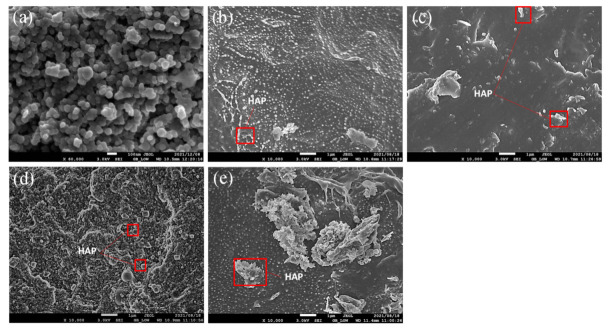
SEM micrographs of (**a**) HAP, (**b**) PCL 1, (**c**) PCL 3, (**d**) PCL 5, and (**e**) PCL 7.

**Figure 2 materials-15-00104-f002:**
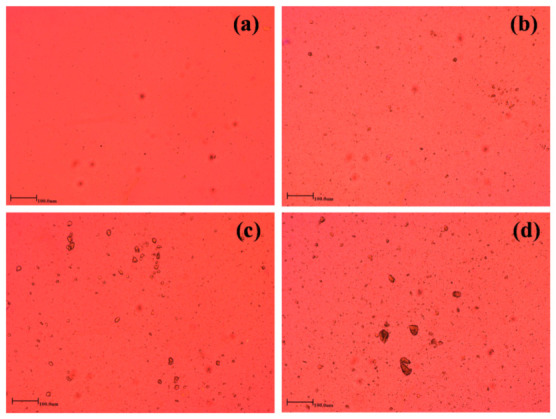
Optical microscopy images of (**a**) PCL 1, (**b**) PCL 3, (**c**) PCL 5, and (**d**) PCL 7 at melt temperature of 100 °C.

**Figure 3 materials-15-00104-f003:**
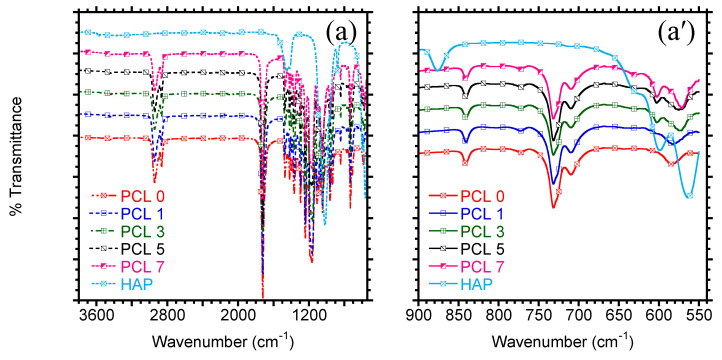
Chemical structures of (**a**) HAP, PCL, and PCL/HAP composites. (**a’**) zoomed region at low wavenumber (550–900 cm^−1^).

**Figure 4 materials-15-00104-f004:**
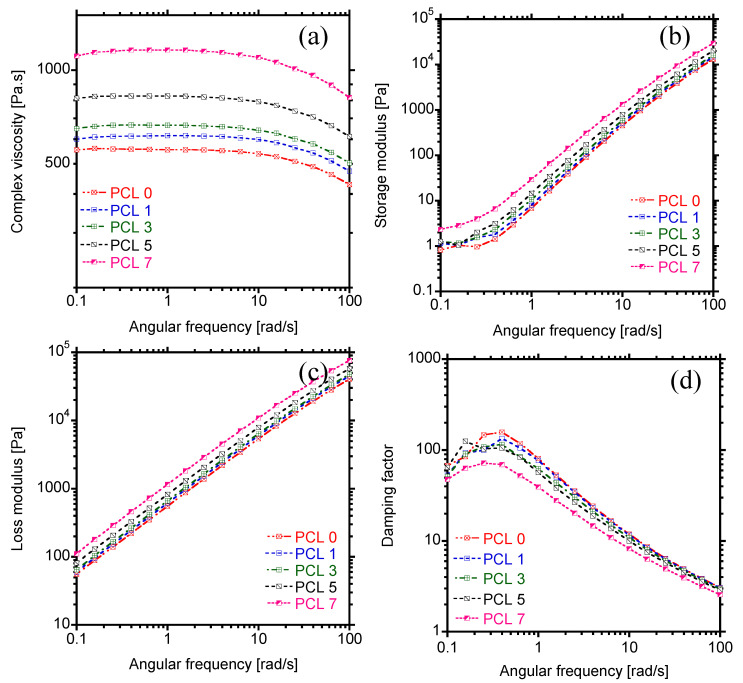
(**a**) Complex viscosity, (**b**) Storage modulus, (**c**) Loss modulus, and (**d**) Damping factor curves of PCL and PCL/HAP composites.

**Figure 5 materials-15-00104-f005:**
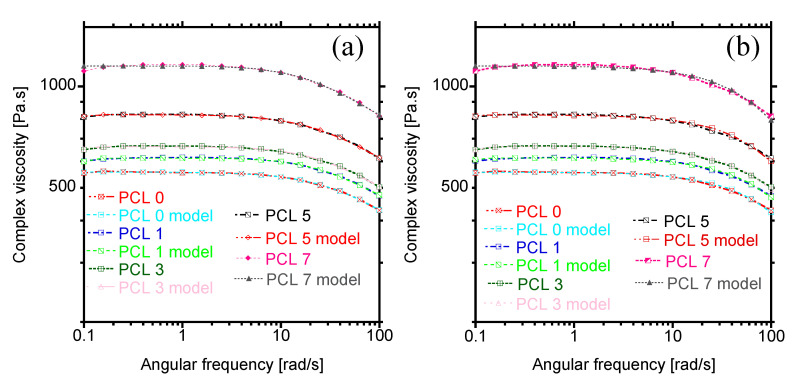
Complex viscosity modelling. (**a**) Carreau–Yasuda and (**b**) Cross model.

**Figure 6 materials-15-00104-f006:**
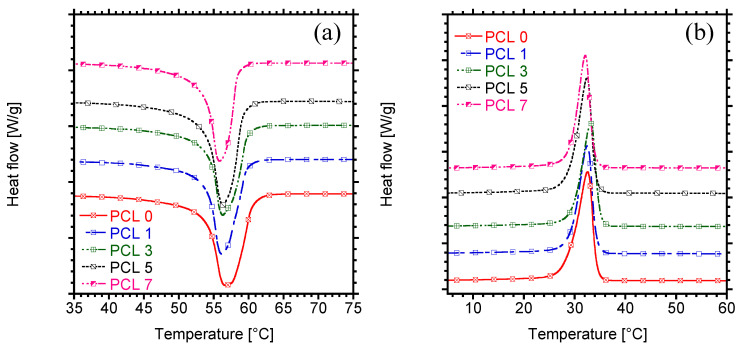
(**a**) DSC melting curves and (**b**) Melt crystallization curves of PCL and PCL/HAP composites.

**Figure 7 materials-15-00104-f007:**
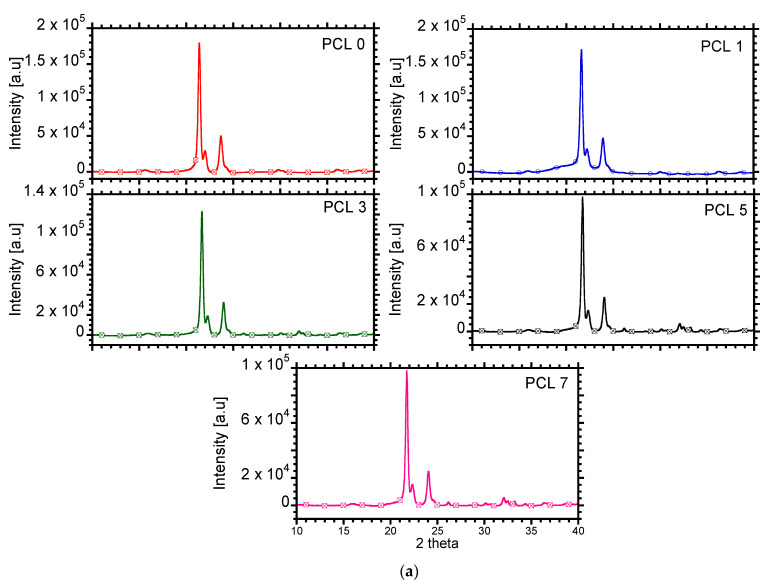
(**a**) XRD diffraction patterns with corrected baseline. (**b**) Change in intensity of (110) and (200) planes as a function of HAP concentration.

**Figure 8 materials-15-00104-f008:**
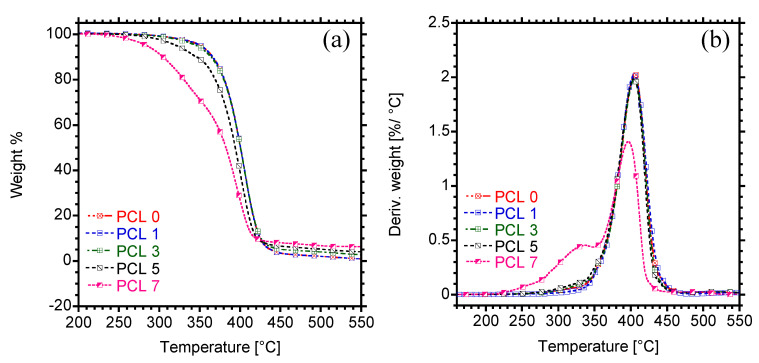
(**a**) Thermal degradation curves and (**b**) Derivative curves of PCL and PCL/HAP composites.

**Figure 9 materials-15-00104-f009:**
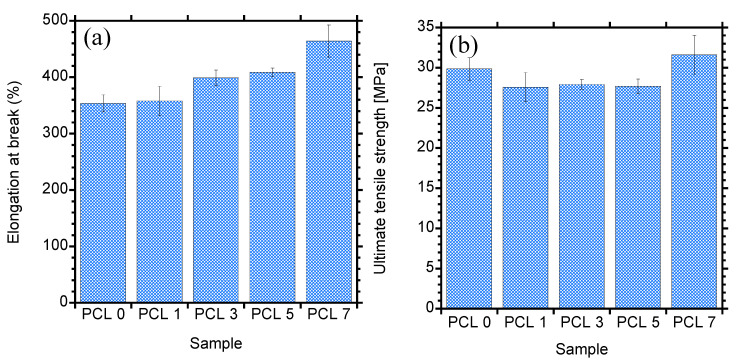
(**a**) Tensile strain and (**b**) ultimate tensile strength curves of PCL and PCL/HAP composites.

**Table 1 materials-15-00104-t001:** Carreau–Yasuda and Cross fitting parameters.

Sample	Carreau–Yasuda Model				Cross Model		
	*η* _0_	*λ*	*n*	*a*	*η* _0_	*λ*	*n*
PCL 0	667.54	0.026	0.818	1.197	556.7	0.003	0.00
PCL 1	1030.3	0.046	0.904	1.681	613.8	0.003	0.02
PCL 3	951.3	0.041	0.873	1.435	604.3	0.003	0.08
PCL 5	1071.2	0.039	0.849	1.299	-	-	-
PCL 7	-	-	-	-	-	-	-

**Table 2 materials-15-00104-t002:** Summarized melting and crystallization temperatures and heat of fusion of PCL and PCL/HAP composites.

Sample	*T_m_* (°C)	Δ*H_m_* (J/g)	*T_c_* (°C)	% Crystallinity
PCL 0	56.7	76.6	31.6	55.1
PCL 1	56.2	76.7	32.5	55.7
PCL 3	56.3	77.5	33.1	57.5
PCL 5	56.4	73.7	32.5	55.8
PCL 7	55.8	69.7	32.1	53.9

**Table 3 materials-15-00104-t003:** Summarized onset and maximum degradation temperatures of PCL and PCL/HAP composites.

Sample	*T* _5%_	*T_max_*
PCL 0	351.3	406.2
PCL 1	360.5	407.1
PCL 3	346.6	405.2
PCL 5	321.5	400.6
PCL 7	284.6	396.9

## Data Availability

Not applicable.

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
