# Peer review of "Viscoelastic, Thermal, and Mechanical Properties of Melt-Processed Poly (ε-Caprolactone) (PCL)/Hydroxyapatite (HAP) Composites"

_materials, 2021, doi:10.3390/ma15010104_

Round 1
Reviewer 1 Report
The manuscript reports the viscoelastic, thermal, and mechanical properties of melt-processed PCL/HAP composites. However, it did not report a significant or enough advancement on PCL/HAP composites. Therefore, it is not suitable for publication in Materials. Nevertheless, further improvements are recommended before resubmitting to other journals.
What is the application direction of PCL/HAP composites? What is the advantage of the work compare to the reported works?
For better understanding Figure 1, the structure and size of HAP should be analyzed by SEM or TEM. And in Figure 1, the author should give which region is HAP.
What is the meaning of PCL1? The author should give the declaration.
The label of Figure 3a’ should be added. All curves of Figure 7a could be drawn in one graph.
The author declared “However, at 5 and 7 wt. % loading, the maximum degradation temperature decreased significantly, possibly because of poor dispersion of HAP in PCL.” (Page 4 line 68) Please provide evidence. Maybe the TG of pure HAP should be given.
In its current state, the level of English throughout the manuscript needs language polishing. Please check the manuscript and refine the language carefully.
Reviewer 2 Report
This manuscript has a systematic arrangement and presents scientific discussion. There are comments to authors that may support and attain the research objectives.
- The conclusion may add the significance of the complex viscosity for revealing the benefit of the melt processing for PCL/HAP composites.
- Please example applications of these composites.
- Please inform an average particle size of HAP (if applicable) and material compositions in the experimental part.
Round 2
Reviewer 1 Report
The authors have made substantial revisions on the manuscript according to the comments from the reviewers. My questions are well addressed and i'd like to recommend the manuscript to publish.